# Women infertility and common mental disorders: A cross-sectional study from North India

Navjot Kamboj[1], Kallur Nava Saraswathy[1], Sweta Prasad[1], Nandita Babu[2], Manju Puri[3], Apoorva Sharma[1], Sukriti Dhingra[4], Mohinder Pal Sachdeva[1], Chakraverti Mahajan[1] *

1 Department of Anthropology, University of Delhi, New Delhi, India, 2 Department of Psychology, University of Delhi, New Delhi, India, 3 Department of Obstetrics and Gynaecology, Lady Hardinge Medical College, Delhi, India, 4 Rehabilitation Council of India, New Delhi, India

* cmahajan@anthro.du.ac.in

**Data Availability Statement:** Data cannot be shared publicly because of ethical issues. Data are available from the Department of Anthropology, University of Delhi Institutional Data Access /

## Abstract

### Background

Infertility is a very distressing condition. It is often associated with long-term stress, which can emerge as anxiety and depression.

### Aim

To understand the effect of socio-demographic variables, reproductive trajectories, and lifestyle variables on stress, depression, and anxiety independently and to understand the relationship of psychological variables with each other among infertile and fertile women.

### Methods

This cross-sectional study recruited 500 women which included 250 primary infertile cases and 250 age-matched fertile controls of the age group 22–35 years. A pretested modified interview schedule was administered which included demographic variables, lifestyle variables, and reproductive trajectories. In addition, psychological tools like PSS, GAD-7, and PHQ-9 were used to collect the data pertaining to Stress, anxiety, and depression, respectively. Data analysis was performed with the statistical software version SPSS, IBM version 24.

### Results

Infertile women are more prone to various psychological disorder (stress, anxiety and depression). None of the demographic and lifestyle variables were associated with stress, anxiety, and depression among infertile women. Only reproductive trajectories were found to be causing stress, anxiety, and depression respectively among infertile women. In addition, stress is leading to both anxiety and depression among infertile women but only to depression in fertile women.

Ethics Committee (contact via cmahajan@anthro.
du.ac.in) for researchers who meet the criteria for
access to confidential data.

**Funding:** The project is funded by the National
Commission for women, Government of India
(Sanction Letter No. – F.No. 16(86)/2019-20)/NCW
(RS)) without publication charges. The funders had
no role in study design, data collection and
analysis, decision to publish, or preparation of the
manuscript. Navjot Kamboj the first author
received salary from the project as Project
Assistant for completing the project for two years.
Apoorva Sharma, Sweta Prasad and Sukriti Dingra
were also in the project as Field Investigators for
one year.

**Competing interests:** The authors have declared
that no competing interests exist.

**Abbreviations:** AIDS, Acquired immune deficiency
syndrome; TB, Tuberculosis; PCOS/PCOD,
Polycystic ovarian syndrome or disease.

## Conclusion

Infertile women should be counselled by medical experts regarding reproductive trajectories. Infertile couples should be guided and counselled to incorporate mental health screening and treatment in their routine check-up.

## Background

According to the World Health Organisation "Infertility is a condition of the male or female reproductive system described by the inability to conceive after 12 months or more of unprotected sexual activity" [1]. Infertility is a chronic struggle for couples and it affects about 10 to 12% of couples worldwide [2]. Infertility in women can be caused by many different factors such as a damaged or blocked fallopian tube, endometriosis, PCOS, and hormonal imbalance [3]. It may work as a painful emotional experience and cause a lot of psychological issues or mental disorders such as stress, anxiety, depression, diminished self-esteem, and reduced quality of life [4]. Studies revealed that most infertile women were found to have higher levels of stress, anxiety, and depression [5, 6], and these disorders were found to be more associated with infertility as compared to other mental disorders [7].

Stress is a physical and emotional reaction that people have when they are confronted with life upheavals. Long time stress can cause a variety of health concerns including depression, anxiety, and other mental illnesses [8]. The association between stress and infertility has been debated for years [4]. Stress can affect fertility in women by suppressing luteinizing hormone, increasing serum cortisol levels preventing implantation of a fertilized egg, and reducing egg quality [9]. If the stress is continued and prolonged it can result in anxiety and depression [6]. Anxiety and depressive disorders are bidirectional risk factors for one another [10]. Various studies concluded that anxiety leads to depression and thus it is a unidirectional phenomenon [11–13], whereas others reported that depression also led to anxiety suggesting its bidirectional mechanism [14, 15].

A Study by Magalhaes et al. (2010) has revealed that the corticotropin-releasing factor receptor 1(CRFR1) and serotonin receptor (5-HTR) are related to anxiety and depression respectively. CRFR1 works to increase the number of 5-HTRs on the cell surface in the brain, which can cause abnormal brain signaling [16]. Since CRFR1 activation leads to anxiety in response to stress and 5-HTRs leads to depression. hence, one can assume that biological stress leads to anxiety which in turn leads to depression.

The relationship between psychological factors (stress, anxiety, and depression) and infertility is rather complex, multifactorial, and bidirectional [17]. Thus, the present study aims to capture the prevalence of stress, depression, and anxiety among infertile and fertile women and to understand the relationship of psychological variables with each other. Attempt was also made to understand the effect of socio-demographic variables, reproductive trajectories, and lifestyle variables on stress, depression, and anxiety.

## Methods

The present study is a part of a Project titled "Infertility and its psychological impact: A case-control study from North India" funded by the National Commission for Women of India, Government of India. Data was collected from 500 females, including 250 infertile and 250 age-matched fertile married women with the age group of 22–35 years from August 2020 to February 2021 after obtaining the ethical clearance from the Department of Anthropology and

Lady Hardinge Medical college & its associated Hospital, New Delhi. After taking a written informed consent the data of infertile women was collected from the Gynaecology Outpatient Department (OPD) of Lady Hardinge Medical College and Smt. Sucheta Kriplani Hospital, New Delhi, and data of fertile women was collected using a household survey in Delhi. The inclusion criteria for infertile women were women seeking infertility treatment and belonging to North India. The inclusion criteria for fertile women were women of the same reproductive age, with successful pregnancy outcomes having children older than one year and also belonging to North India.

A pretested modified interview schedule was used to collect data pertaining to demographic variables (age, educational status, occupational status, social-economic status [18] and family structure), lifestyle variables (sleep pattern, physical activity, exercise), and reproductive trajectories (age at menarche, age at marriage, bleeding during menstruation duration of infertility, adoption of birth control pills, intercourse during fertile period, possible causes of infertility). The psychological tools like Perceived Stress Scale (PSS), Generalized Anxiety Disorder (GAD-7) and Patient Health Questionnaire (PHQ-9) were used to collect the data pertaining to stress, anxiety, and depression, respectively.

## Statistical analysis

In the present study, as the distribution of data was found to be skewed, so median scores of all psychological variables were considered and Mann- Whitney test was performed to compare the groups. Further, Pearson's chi-square test was used to compare the distribution of stress, anxiety, and depression among infertile and fertile women. In order to identify the causes of stress, anxiety, and depression among infertile and fertile women linear regression analysis was done. Data analysis was performed with the statistical software SPSS, IBM version 24, and the significance level for all the data was <0.05.

## Results

### Median of stress, depression, and anxiety scores of infertile and fertile women

The median scores compared between the fertile and infertile women revealed that the median levels of stress and anxiety were significantly higher among infertile women as compared to that of fertile women (p<0.05). However, the median levels of depression among the fertile and infertile women were found to be similar (p >0.05) (Table 1).

### Stress, anxiety, and depression among infertile and fertile women

The infertile and fertile women were compared in terms of stress, anxiety, and depression with reference to distribution of frequency, which revealed that the individuals with high stress, anxiety, and depression were found to be significantly higher among the infertile women as compared to fertile women (p<0.05) (Table 2).

### Distribution of demographic variables, lifestyle variables, and reproductive trajectories among infertile and fertile women on the variables of stress, depression, and anxiety

Further, the difference between non-stressed (low stress) and stressed (medium and high stress), anxious (moderately severe and severe anxiety) and non-anxious (mild and moderate anxiety), and depressed (mild, moderate, moderately severe, and severe depression) and non-depressed (normal) women were compared with respect to demographic, lifestyle variables, and reproductive trajectories, independently in both infertile and fertile women. The duration

**Table 1. Median scores of stress, depression and anxiety among infertile and fertile women.**

| Psychological Variables | Infertile Median (IQR) | Fertile Median (IQR) | Mann Whitney Test P-value |
|---|---|---|---|
| Stress (PSS) | 21(16–26) | 17 (12–21) | 0.001 |
| Anxiety (GAD-7) | 8(5–12) | 4 (3–9) | 0.001 |
| Depression (PHQ-9) | 9 (5–13) | 8 (5–12) | 0.49 |

IQR- Interquartile range, Level of significance p<0.0.05

of infertility and possible causes of infertility in low versus highly stressed, anxious, and depressed infertile women were also compared (Tables 3–5).

## Demographic variables, lifestyle variables, and reproductive trajectories with reference to stress

The results indicate that infertile women whose husbands were having private jobs or were daily wage earners were found to be more stressed as compared to those who were unemployed or were either having government jobs or their own business (p = 0.001). The infertile women who rarely exercised were observed to be more stressed as compared to those who exercised frequently (p = 0.006). The women who had infrequent intercourse during the fertile period were found to be more stressed in comparison to those who had frequent intercourse during the fertile period among the infertile as well as fertile women category (p = 0.01). In the case of fertile women, those who were extremely active were found to be more stressed as compared to those who were less active or not active at all (p = 0.003).

## Demographic variables, lifestyle variables, and reproductive trajectories with reference to anxiety

Among the infertile women, those who were illiterate or were having primary education were found to be more anxious as compared to those who had higher education (p = 0.03). The infertile women who were slightly active or not active were more anxious as compared to those who were extremely or moderately active (p = 0.003). The infertile women who got married before the age of 18 years were found to be more anxious as compared to those who got

**Table 2. Distribution of stress, anxiety, and depression among infertile and fertile women.**

| Psychological Variables | Levels (Scores) | Infertile (250) N (%) | Fertile (250) N (%) | (P- Value) |
|---|---|---|---|---|
| Stress | Low (0–13) | 33(13.2) | 88(35.2) | 0.001 |
| | Medium (14–26) | 160(64.0) | 152(60.8) | |
| | High (27–40) | 57(22.8) | 10(4.0) | |
| Anxiety | Mild (0–4) | 59(23.6) | 189(75.6) | 0.001 |
| | Moderate (5–9) | 82(32.8) | 52(20.8) | |
| | Moderately Severe (10–15) | 69(27.6) | 6(2.4) | |
| | Severe (15–21) | 40(16.0) | 3(1.2) | |
| Depression | Normal(0–4) | 58(23.2) | 191(76.4) | 0.001 |
| | Mild (5–9) | 82(32.8) | 29(11.6) | |
| | Moderate(10–14) | 65(26.0) | 23(9.2) | |
| | Moderately severe(15–19) | 38(15.2) | 7(2.8) | |
| | Severe (20–27) | 7(2.8) | 0(0) | |

N is the sample size, Level of significance p<0.05

**Table 3. Distribution of demographic, lifestyle variables and reproductive trajectories in infertile and fertile women in stress category.**

| CHARACTERISTICS | | INFERTILE WOMEN | | P-value | FERTILE WOMEN | | P-value |
|---|---|---|---|---|---|---|---|
| | | Stressed (217) N(%) | Non-stressed (33) N(%) | | Stressed (162) N(%) | Non-stressed (88) N(%) | |
| Age (years) | Upto 25 | 84 (38.7) | 16 (48.48) | 0.52 | 63 (38.89) | 36 (40.9) | 0.87 |
| | 26–30 | 95 (43.78) | 13 (39.39) | | 71 (43.83) | 39 (44.32) | |
| | 31–35 | 38 (17.52) | 4 (12.12) | | 28 (17.28) | 13 (14.78) | |
| Educational status | illiterate/informal | 27(12.44) | 1 (3.03) | 0.28 | 58 (35.8) | 23 (26.14) | 0.6 |
| | Primary | 22 (10.14) | 1 (3.03) | | 63 (38.89) | 38 (43.18) | |
| | Middle | 14 (6.45) | 2 (6.06) | | 27 (16.67) | 19 (21.59) | |
| | secondary/senior secondary | 41 (18.89) | 7 (21.21) | | 6 (3.70) | 3 (3.41) | |
| | Higher | 113(52.08) | 22 (66.66) | | 8 (4.94) | 5 (5.68) | |
| Family Structure | Nuclear | 96 (44.24) | 14 (42.42) | 0.85 | 126 (77.78) | 59 (67.05) | 0.06 |
| | Joint | 121 (55.76) | 19 (57.58) | | 36 (22.22) | 29 (32.95) | |
| Occupational status | Not working | 3 (1.38) | 2 (6.06) | **0.001** | 3 (1.85) | 2 (2.27) | 0.08 |
| | daily wage earner | 68 (31.34) | 2 (6.06) | | 105 (64.81) | 57 (64.77) | |
| | private job | 88 (40.55) | 11 (33.33) | | 31 (19.13) | 25 (28.41) | |
| | govt job | 25 (11.52) | 9 (27.27) | | 15 (9.28) | 4 (4.55) | |
| | Business | 33 (15.21) | 9 (27.27) | | 8 (4.94) | 0 (0) | |
| Family Income | Upper lower | 44 (20.29) | 3 (9.09) | 0.4 | 50 (30.86) | 30 (34.09) | 0.85 |
| | Lower middle | 49 (22.59) | 8 (24.24) | | 28 (17.29) | 13 (14.78) | |
| | Upper middle | 124 (57.3) | 22 (66.66) | | 84 (51.9) | 45 (51.1) | |
| Physical Activity | Extremely active | 34 (15.67) | 10 (30.30) | 0.24 | 68 (41.97) | 22 (25.00) | **0.003** |
| | Moderately active | 80 (36.86) | 11 (33.33) | | 32 (19.75) | 20 (22.73) | |
| | Active | 55 (25.35) | 8 (24.24) | | 52 (32.1) | 43 (48.86) | |
| | Slightly active | 39 (17.97) | 4 (12.12) | | 10 (6.18) | 1 (1.14) | |
| | Not active at all | 9 (4.15) | 0 (0) | | 0 (0) | 2 (2.27) | |
| Exercise | Never | 100 (46.08) | 19 (57.58) | **0.006** | 59 (36.42) | 36 (40.91) | 0.53 |
| | Seldom | 29 (13.36) | 4 (12.12) | | 72 (44.44) | 42 (47.73) | |
| | Sometimes | 60 (27.65) | 1 (3.03) | | 29 (17.91) | 10 (11.36) | |
| | Often | 9 (4.15) | 1 (3.03) | | 2 (1.23) | 0 (0) | |
| | Very often | 19 (8.76) | 8 (24.24) | | 0 (0) | 0 (0) | |
| Sleep pattern(hours) | <6 | 31 (14.29) | 6 (18.18) | 0.42 | 24 (14.81) | 8 (9.09) | 0.54 |
| | 6–7 | 86 (39.63) | 9 (27.27) | | 70 (43.21) | 38 (43.19) | |
| | 7–8 | 73 (33.64) | 15 (45.45) | | 65 (40.12) | 41 (46.59) | |
| | >8 | 27 (12.44) | 3 (9.09) | | 3 (1.86) | 1 (1.14) | |
| Age of menarche(years) | 9–12 | 55 (25.36) | 6 (18.18) | 0.31 | 43 (26.54) | 29 (32.96) | 0.52 |
| | 13–16 | 154 (70.95) | 27 (81.82) | | 118 (72.84) | 58 (65.9) | |
| | Above 16 | 8 (3.69) | 0 (0) | | 1 (0.62) | 1 (1.14) | |
| Disturbed menstruation | Yes | 85 (39.17) | 12 (36.36) | 0.76 | 5 (3.09) | 0 (0) | 0.09 |
| | No | 132 (60.83) | 21 (63.64) | | 157 (96.91) | 88 (100) | |
| Age of marriage(years) | 10–17 | 21 (9.68) | 1 (3.03) | 0.24 | 9 (5.55) | 7 (7.96) | 0.42 |
| | 18–25 | 149 (68.67) | 28 (84.85) | | 145 (89.51) | 80 (90.09) | |
| | 26–32 | 46 (21.19) | 4 (12.12) | | 8 (4.94) | 1 (1.14) | |
| | Above 32 | 1 (0.46) | 0 (0) | | 0 (0) | 0 (0) | |
| Bleeding during menstruation | Heavy bleeding | 36 (16.59) | 5 (15.15) | 0.97 | 56 (34.57) | 39 (44.32) | 0.31 |
| | Normal bleeding | 149 (68.66) | 23 (69.70) | | 85 (52.47) | 40 (45.45) | |
| | Less bleeding | 32 (14.75) | 5 (15.15) | | 21 (12.96) | 9 (10.23) | |

*(Continued)*

**Table 3.** (Continued)

| CHARACTERISTICS | | INFERTILE WOMEN | | P-value | FERTILE WOMEN | | P-value |
|---|---|---|---|---|---|---|---|
| | | Stressed (217) N(%) | Non-stressed (33) N(%) | | Stressed (162) N(%) | Non-stressed (88) N(%) | |
| Duration of infertility(years) | 1–5 | 152 (70.04) | 30 (90.91) | 0.08 | - | - | |
| | 6–10 | 46 (21.20) | 3 (9.09) | | - | - | |
| | 11–15 | 16 (7.37) | 0 (0) | | - | - | |
| | Above 15 | 3 (1.39) | 0 (0) | | - | - | |
| Adoption of birth control without consulting doctors | Yes | 16 (7.37) | 4 (12.12) | 0.34 | 35 (21.6) | 20 (22.73) | 0.84 |
| | No | 201 (92.63) | 29 (87.88) | | 127 (78.4) | 68 (77.27) | |
| Intercourse during fertile period | Always | 85 (39.17) | 21 (63.64) | **0.01** | 1 (0.62) | 4 (4.54) | **0.04** |
| | Sometimes | 98 (45.16) | 6 (18.18) | | 88 (54.32) | 54 (61.37) | |
| | Never | 34 (15.67) | 6 (18.18) | | 73 (45.06) | 30 (34.09) | |
| Possible Causes of infertility | Blocked tubes | 29 (13.3) | 11 (33.34) | 0.36 | - | - | |
| | Irregular menstruation | 39 (17.90) | 5 (15.15) | | - | - | |
| | PCOS/PCOD | 11 (5.06) | 1 (3.03) | | - | - | |
| | Cyst | 28 (12.90) | 2 (6.06) | | - | - | |
| | Endometriosis, Mass development uterus | 11 (5.08) | 1 (3.03) | | - | - | |
| | Fibroid | 10 (4.60) | 2 (6.06) | | - | - | |
| | Hormonal imbalance | 4 (1.86) | 0 (0) | | - | - | |
| | No menstruation | 9 (4.14) | 1 (3.03) | | - | - | |
| | Thyroid/AIDS / TB | 28 (12.90) | 4 (12.12) | | - | - | |
| | Other reasons* | 48 (22.14) | 6 (18.18) | | - | - | |

N = number of individuals, Level of significance p < 0.05.

* Uterine infection, Salpingectomy and Tubal surgery

married after 18 years of age (p = 0.02). The women who were infertile for more than 5 years were found to be more anxious in comparison to those who were infertile for less than 5 years (p = 0.02). Among the fertile women, those women who were below 25 years of age were found to be more anxious in comparison to those who were older than 25 years (p = 0.007). The women who were having disturbed menstruation were found to be more anxious as compared to those having regular menses (p = 0.000). The fertile women who were having heavy bleeding during menstruation were found to be more anxious compared to those who were either having less bleeding or were having normal bleeding during menstruation (p = 0.04).

## Demographic variables, lifestyle variables, and reproductive trajectories with reference to depression

Among the infertile women, those who never exercised were found to be more depressed as compared to those who sometimes or often exercised (p = 0.008). The infertile women who were taking more than 8 hours of sleep were more depressed as compared to those who were taking less than 8 hours of sleep (p = 0.009). The women having less bleeding during menstruation were found to be more depressed as compared to those who were having heavy bleeding during menstruation in the infertile category (p = 0.00). In the case of fertile women, those who were living in a joint family were more depressed as compared to those living in a nuclear family (p = 0.009). The fertile women who sometimes exercised were more depressed as compared to those who never exercised or exercised regularly (p = 0.01). Those fertile women who

**Table 4. Distribution of demographic, lifestyle variables and reproductive trajectories in infertile and fertile women in the anxiety category.**

| CHARACTERISTICS | | INFERTILE WOMEN | | P-value | FERTILE WOMEN | | p-value |
|---|---|---|---|---|---|---|---|
| | | Anxious (109) N(%) | Non-anxious (141) N(%) | | Anxious (9) N (%) | Non-anxious (241) N(%) | |
| Age(years) | Upto 25 | 45 (41.28) | 55 (39.01) | 0.70 | 8 (88.89) | 91 (37.76) | **0.007** |
| | 26–30 | 44 (40.37) | 64 (45.39) | | 0 (0) | 110 (45.64) | |
| | 31–35 | 20 (18.35) | 22 (15.60) | | 1 (11.11) | 40 (16.60) | |
| Educational status | illiterate/informal | 15 (13.76) | 13 (9.22) | **0.03** | 5 (55.55) | 76 (31.54) | 0.6 |
| | Primary | 16 (14.68) | 7 (4.96) | | 3 (33.33) | 98 (40.66) | |
| | Middle | 8 (7.34) | 8 (5.68) | | 1 (11.11) | 45 (18.67) | |
| | secondary/senior secondary | 21 (19.27) | 27 (19.15) | | 0 (0) | 9 (3.73) | |
| | Higher | 49 (44.95) | 86 (60.99) | | 0 (0) | 13 (5.59) | |
| Family Structure | Nuclear | 48 (44.04) | 62 (43.97) | 1 | 8 (88.89) | 177 (73.44) | 0.3 |
| | Joint | 61 (55.96) | 79 (56.03) | | 1 (11.11) | 64 (26.56) | |
| Occupational status | Not working | 2 (1.83) | 3 (2.13) | 0.90 | 0 (0) | 5 (2.07) | 0.63 |
| | daily wage earner | 34 (31.19) | 36 (25.53) | | 8 (88.89) | 154 (63.9) | |
| | private job | 42 (38.53) | 57 (40.43) | | 1 (11.11) | 55 (22.82) | |
| | Govt. job | 14 (12.84) | 20 (14.18) | | 0 (0) | 19 (7.88) | |
| | Business | 17 (15.60) | 25 (17.73) | | 0 (0) | 8 (3.32) | |
| Family Income | Upper lower | 26 (23.86) | 21 (14.89) | 0.19 | 3 (33.33) | 77 (31.95) | 0.86 |
| | Lower middle | 24 (22.03) | 33 (23.40) | | 2 (22.22) | 39 (16.18) | |
| | Upper middle | 59 (54.2) | 87(61.7) | | 4(44.44) | 125 (51.8) | |
| Physical activity | Extremely active | 18 (16.51) | 26 (18.44) | **0.003** | 4 (44.44) | 86 (35.68) | 0.09 |
| | Moderately active | 33 (30.26) | 58 (41.13) | | 1 (11.11) | 51 (21.16) | |
| | Active | 23 (21.10) | 40 (28.37) | | 2 (22.22) | 93 (38.59) | |
| | Slightly active | 28 (25.67) | 15 (10.64) | | 2 (22.22) | 9 (3.73) | |
| | Not active at all | 7 (6.4) | 2 (1.42) | | 0 (0) | 2 (0.83) | |
| Exercise | Never | 48 (44.04) | 71 (50.35) | 0.08 | 3 (33.33) | 92 (38.17) | 0.65 |
| | Seldom | 20 (18.35) | 13 (9.22) | | 6 (66.67) | 108 (44.81) | |
| | Sometimes | 21 (19.27) | 40 (28.37) | | 0 (0) | 39 (16.18) | |
| | Often | 6 (5.5) | 4 (2.84) | | 0 (0) | 2 (0.83) | |
| | Very often | 14 (12.84) | 13 (9.22) | | 0 (0) | 0 (0) | |
| Sleep pattern(hours) | <6 | 20 (18.35) | 17 (12.06) | 0.22 | 1 (11.11) | 31 (12.86) | 0.86 |
| | 6–7 | 34 (31.19) | 61 (43.26) | | 3 (33.33) | 105 (43.57) | |
| | 7–8 | 41 (37.61) | 47 (33.33) | | 5 (55.56) | 101 (41.91) | |
| | >8 | 14 (12.84) | 16 (11.35) | | 0 (0) | 4 (1.66) | |
| Age of menarche(years) | 9–12 | 25 (22.94) | 36 (25.53) | 0.85 | 3 (33.33) | 69 (28.63) | 0.92 |
| | 13–16 | 80(73.39) | 101 (70.92) | | 6 (6.67) | 170 (70.54) | |
| | Above 16 | 4 (3.67) | 4 (2.84) | | 0 (0) | 2 (0.83) | |
| Disturbed menstruation | Yes | 48 (44.04) | 49 (34.75) | 0.14 | 2 (22.22) | 3 (1.24) | **0.000** |
| | No | 61 (55.96) | 92 (65.25) | | 7 (77.78) | 238 (98.76) | |
| Age of marriage(years) | 10–17 | 16 (14.68) | 6 (4.26) | **0.02** | 1 (11.11) | 15 (6.22) | 0.88 |
| | 18–25 | 72 (66.05) | 105 (74.46) | | 8 (88.89) | 217 (90.04) | |
| | 26–32 | 21 (19.27) | 29 (20.57) | | 0 (0) | 9 (3.73) | |
| | 32 above | 0 (0) | 1 (0.71) | | 0 (0) | 0 (0) | |
| Bleeding during menstruation | Heavy bleeding | 23 (21.01) | 18 (12.77) | 0.97 | 7 (77.78) | 88 (36.51) | **0.04** |
| | Normal bleeding | 68 (62.39) | 104 (73.05) | | 2 (22.22) | 123 (51.04) | |
| | Less bleeding | 18 (16.51) | 19 (13.48) | | 0 (0) | 30 (12.45) | |

*(Continued)*

**Table 4.** (Continued)

| CHARACTERISTICS | | INFERTILE WOMEN | | P-value | FERTILE WOMEN | | p-value |
|---|---|---|---|---|---|---|---|
| | | Anxious (109) N(%) | Non-anxious (141) N(%) | | Anxious (9) N (%) | Non-anxious (241) N(%) | |
| Duration of infertility | 1–5 | 70 (64.22) | 112 (79.43) | **0.02** | - | - | |
| | 6–10 | 27 (24.77) | 22 (15.6) | | - | - | |
| | 11–15 | 9 (8.26) | 7 (4.96) | | - | - | |
| | Above 15 | 3 (2.75) | 0 (0) | | - | - | |
| Adoption of birth control without consulting doctors | Yes | 11 (10.09) | 9 (6.38) | 0.28 | 3 (33.33) | 52 (21.58) | 0.42 |
| | No | 98 (89.91) | 132 (93.62) | | 6 (66.67) | 189 (78.42) | |
| Intercourse during fertile period | Always | 40 (36.70) | 66 (46.81) | 0.20 | 0 (0) | 5 (2.07) | 0.42 |
| | Sometimes | 52 (47.70) | 52 (36.88) | | 7 (77.78) | 135 (56.02) | |
| | Never | 17 (15.60) | 23 (16.31) | | 2 (22.22) | 101 (41.91) | |
| Possible Causes of infertility | Blocked tubes | 14 (12.84) | 26 (18.44) | 0.17 | - | - | |
| | Irregular menstruation | 27 (24.77) | 17 (12.06) | | - | - | |
| | PCOS/PCOD | 4 (3.67) | 8 (5.67) | | - | - | |
| | Cyst | 16 (14.68) | 14 (9.93) | | - | - | |
| | Endometriosis, Mass development uterus | 4 (3.67) | 8 (5.67) | | - | - | |
| | Fibroid | 3 (2.75) | 9 (6.38) | | - | - | |
| | Hormonal imbalance | 2 (1.83) | 2 (1.42) | | - | - | |
| | No menstruation | 6 (5.5) | 4 (2.84) | | - | - | |
| | Thyroid/ / AIDS/ TB | 12 (11.01) | 20 (14.18) | | - | - | |
| | Other reasons* | 21 (19.28) | 33 (23.4) | | - | - | |

N = number of individuals, Level of significance p < 0.05.

* Uterine infection, Salpingectomy and Tubal surgery.

were taking more than 7 hours of sleep daily were more depressed as compared to those taking below 7 hours of sleep (p = 0.001). Among the fertile women, those who got married early i.e., before 18 years of age were observed to be more depressed as compared to those who got married after 18 years of age (p = 0.005). Lastly, the fertile women who were having normal bleeding during menstruation were found to be more depressed as compared to those having less or heavy bleeding (p = 0.03).

Further, many other variables were similarly distributed among depressed and non-depressed women in both infertile and fertile category i.e., among the infertile women, those who were below 25 years of age, those who were illiterate, those who were living in a joint family, those husbands were having private jobs, those belonging to upper lower socioeconomic status, those who were slightly active or were not active at all, women whose age at menarche was 13–16 years, those who had disturbed menstruation, those who got married before 18 years of age, those who took birth control pills in the past and those having irregular menstruation and other reasons as their possible causes of infertility were more depressed as compared to other subgroups of the respective variables (p> 0.05).

## Association of statistically significant demographic variables, lifestyle variables, and reproductive trajectories with the psychological variables

Linear regression was performed to understand the association between various demographic variables, lifestyle variables, and reproductive trajectories which were significantly different

**Table 5. Distribution of demographic, lifestyle variables and reproductive trajectories in infertile and fertile women in depression category.**

| CHARACTERISTICS | | INFERTILE WOMEN | | P-value | FERTILE WOMEN | | P-value |
|---|---|---|---|---|---|---|---|
| | | Depressed 192 N (%) | Normal 58 N (%) | | Depressed 59 N (%) | Normal 191 N (%) | |
| Age(years) | Upto 25 | 82(42.7) | 17(29.3) | 0.17 | 21(35.6) | 78(40.8) | 0.22 |
| | 25–30 | 81(42.2) | 29(50) | | 24(40.7) | 86(45) | |
| | 31–35 | 29(15.1) | 12(20.7) | | 14(23.7) | 27(14.1) | |
| Educational status | Illiterate/informal | 24(12.5) | 4(6.9) | 0.69 | 17(28.8) | 64(33.5) | 0.18 |
| | Primary | 17(8.85) | 6(10.34) | | 19(32.2) | 82(42.9) | |
| | Middle | 13(6.77) | 3(5.17) | | 15(25.4) | 31(16.2) | |
| | Secondary/Senior secondary | 38(19.79) | 10(17.24) | | 4(6.8) | 5(2.6) | |
| | Higher | 100(52.08) | 35(60.34) | | 4(6.8) | 9(4.7) | |
| Family Structure | Nuclear | 83(43.23) | 27(46.6) | 0.65 | 36(61) | 149(78) | **0.009** |
| | Joint | 109(56.77) | 31(53.4) | | 23(39) | 42(22) | |
| Occupational status | Not working | 4(2.08) | 0(0) | 0.30 | 2(3.4) | 3(1.6) | 0.36 |
| | Daily wage earner | 53(27.60) | 17(29.3) | | 43(72.9) | 119(62.3) | |
| | Private job | 80(41.7) | 18(31) | | 9(15.3) | 47(24.6) | |
| | Govt job | 24(12.5) | 12(20.7) | | 3(5.1) | 16(8.4) | |
| | Business | 31(16.15) | 11(19) | | 2(3.4) | 6(3.1) | |
| Family Income | Upper lower | 39(20.31) | 8(13.8) | 0.29 | 16(27.1) | 64(33.5) | 0.05 |
| | Lower middle | 40(20.83) | 17(29.3) | | 5(8.5) | 36(18.8) | |
| | Upper middle | 113(58.8) | 33(56.8) | | 38(64.4) | 91(47.6) | |
| Physical activity | Extremely active | 32(16.7) | 12(20.7) | 0.38 | 27(45.8) | 64(33.5) | 0.32 |
| | Moderately active | 67(34.9) | 24(41.4) | | 10(16.9) | 43(22.5) | |
| | Active | 49(25.5) | 14(24.1) | | 18(30.5) | 77(40.3) | |
| | Slightly active | 35(18.2) | 8(13.8) | | 4(6.8) | 7(3.7) | |
| | Not active at all | 9(4.7) | 0(0) | | 0(0) | 0(0) | |
| Exercise | Never | 100(52.1) | 19(32.8) | **0.008*** | 18(30.5) | 77(40.3) | **0.01*** |
| | Seldom | 25(13.0) | 8(13.8) | | 24(40.7) | 90(47.1) | |
| | Sometimes | 37(19.3) | 24(41.4) | | 15(25.4) | 24(12.6) | |
| | Often | 7(3.6) | 3(5.2) | | 2(3.4) | 0(0) | |
| | Most often | 23(12) | 4(6.9) | | 0(0) | 0(0) | |
| Sleep pattern(hours) | <6 | 31(16.15) | 6(10.3) | **0.009*** | 10(16.9) | 22(11.5) | **0.001*** |
| | 6–7 | 66(34.35) | 29(50) | | 14(23.7) | 94(49.2) | |
| | 7–8 | 68(35.42) | 20(34.5) | | 32(54.2) | 74(38.7) | |
| | >8 | 27(14.06) | 3(5.2) | | 3(5.1) | 1(0.5) | |
| Age of menarche(years) | 9–12 | 46(24) | 15(25.9) | 0.82 | 24(40.7) | 48(25.1) | 0.06 |
| | 13–16 | 139(72.4) | 40(69) | | 35(59.3) | 141(73.8) | |
| | Above 16 | 7(3.7) | 3(5.2) | | 0(0) | 2(1.0) | |
| Disturbed menstruation | Yes | 76(39.6) | 21(36.2) | 0.64 | 4(6.8) | 1(0.5) | 0.003*** |
| | No | 116(60.4) | 37(63.8) | | 55(93.2) | 190(99.5) | |
| Age of marriage (years) | 10–17 | 21(10.9) | 3(5.2) | 0.13 | 9(15.3) | 7(3.7) | **0.005*** |
| | 18–25 | 133(69.3) | 39(67.2) | | 46(78) | 179(93.7) | |
| | 25–32 | 38(19.8) | 15(25.9) | | 4(6.8) | 5(2.6) | |
| | 32 above | 0(0) | 1(1.7) | | 0(0) | 0(0) | |
| Adoption of birth control without consulting doctors | Yes | 16(8.3) | 4(6.9) | 0.13 | 10(16.9) | 44(23) | 0.35 |
| | No | 176(91.7) | 54(93.1) | | 48(81.4) | 147(77) | |

*(Continued)*

**Table 5.** (Continued)

| CHARACTERISTICS | | INFERTILE WOMEN | | P-value | FERTILE WOMEN | | P-value |
|---|---|---|---|---|---|---|---|
| | | Depressed 192 N (%) | Normal 58 N (%) | | Depressed 59 N (%) | Normal 191 N (%) | |
| Bleeding during menstruation | Heavy bleeding | 10(5.2) | 31(53.4) | 0.00* | 21(35.6) | 74(38.7) | 0.03* |
| | Normal bleeding | 152(79.1) | 18(31.0) | | 36(61.0) | 89(46.6) | |
| | Less bleeding | 30(15.6) | 9(15.5) | | 2(3.4) | 28(14.7) | |
| Intercourse during fertile period | Always | 83(43.2) | 25(43.1) | 0.26 | 2(3.4) | 3(1.6) | 0.02 |
| | Sometimes | 77(40.1) | 28(48.3) | | 42(71.2) | 100(52.4) | |
| | Never | 32(16.7) | 5(8.6) | | 15(25.4) | 88(46.1) | |
| Duration of Infertility(years) | 1–5 | 135(70.3) | 47(81.0) | 0.50 | | | |
| | 6–10 | 42(21.8) | 7(12.0) | | | | |
| | 11–15 | 12(6.25) | 4(6.8) | | | | |
| | Above 15 | 3(1.56) | 0(0) | | | | |
| Possible Causes of infertility | Blocked tubes | 32(16.7) | 8(13.8) | 0.48 | | | |
| | Irregular menstruation | 35(18.2) | 9(15.5) | | | | |
| | PCOS/PCOD | 8(4.2) | 4(6.9) | | | | |
| | Cyst | 24(12.5) | 6(10.3) | | | | |
| | Endometriosis, Mass development uterus | 10(5.2) | 2(3.4) | | | | |
| | Fibroid | 9(4.7) | 3(5.2) | | | | |
| | Hormonal imbalance | 4(2.1) | 0(0) | | | | |
| | No menstruation | 8(4.2) | 2(3.4) | | | | |
| | Thyroid/ / AIDS/ TB | 19(9.9) | 13(22.4) | | | | |
| | Other reasons* | 43(22.4) | 11(19.0) | | | | |

N = number of individuals, Level of significance p < 0.05.

* Uterine infection, Salpingectomy and Tubal surgery.

between stress and non-stressed, anxious and non-anxious, and depressed and non-depressed fertile and infertile women. The results revealed that the infertile women who had intercourse infrequent (sometimes) during the fertile period had significantly increased stress by 2.14 times (p = 0.01) whereas the stress is significantly increased by 0.49 times among fertile women due to extreme physical activity(p = 0.01).

Regarding anxiety, women who are infertile for 6–10 years showed a significant increase in anxiety by 2.10 times (p = 0.003). Heavy bleeding during menstruation increased anxiety by 0.60 times among fertile women (p = 0.04) whereas, less bleeding during menstruation significantly increased depression by 1.21 times among infertile women (p = 0.005). Infrequent (sometimes) intercourse during the fertile period showed a mild decrease in depression among fertile women (p = 0.04) (Table 6).

## Association of stress with depression and anxiety among infertile and fertile women

Finally, linear regression was performed to understand the association between stress and other psychological variables, i.e., Depression and Anxiety, which revealed that among the infertile women, a unit change in stress significantly increased depression by 0.14 times (p = 0.001) and anxiety by 0.10 times (p = 0.02). Whereas, among the fertile women, a unit change in stress significantly increases depression by 0.15 times.(0.001) (Table 7).

**Table 6. Linear regression with respect to statistically significant demographic, lifestyle, and reproductive trajectories among infertile and fertile women.**

| Dependent Variables | Infertile | | | Fertile | | |
|---|---|---|---|---|---|---|
| | Significant (Independent variable) | B coefficient (Standard error) | P-value | Significant (Independent variable) | B coefficient (Standard error) | P-value |
| Stress | Infrequent intercourse during the fertile period | 2.14 (0.899) | **0.01** | Less Physical Activity | 0.496 (0.195) | **0.01** |
| Anxiety | Higher Duration of infertility | 2.10 (0.708) | **0.003** | Heavy Bleeding during menstruation | 0.602 | **0.04** |
| | | | | | 0.296 | |
| Depression | Less Bleeding during menstruation | 1.21 (0.434) | **0.005** | Infrequent Intercourse during fertile period | -0.877 (0.443) | **0.04** |

## Discussion

The findings of the present study revealed that the prevalence of stress, anxiety, and depression is more among infertile women as compared to fertile women. The results are in concordance with the findings of previous studies which also reported an increased level of stress, anxiety [6, 19, 20], and depression associated with infertility [21]. Infertility is a condition where a woman experiences social, biological, and cultural trauma, which might act as external stimuli and in turn are likely to cause stress. Long-term stress is an endophenotype for various psychological dysfunctions including anxiety, depression, and cognitive impairment [22].

Further, the distribution of demographic variables, lifestyle variables, and reproductive trajectories among infertile and fertile women vis-a-vis stress, depression, and anxiety was also investigated. In reference to stress, the infertile women whose husbands were private job employees or were daily wage earners were found to be more stressed. This could be attributed to the work-life imbalance, unsatisfactory job roles, and financial instability in addition to the trauma of infertility [23, 24].

The results also revealed that the infertile women who never exercised were found to be more stressed and more depressed. As observed by the field investigators of the present study, exercise among infertile women is one of the recommendations of the doctors as a part of their treatment protocol specifically for the reduction of PCOS and other related infertility issues. Some intervention and prospective studies have also demonstrated that lesser exercise results in higher levels of perceived stress [25, 26]. Further, clinical trials have demonstrated that exercise is an effective method for reducing perceived stress, stress symptoms, and quality of life [25]. Among the fertile group, physically active women were found to be more stressed, which could be due to workload, the pressure of household chores, or other family or children-related issues [27]. In support of this, linear regression revealed that extreme physical activity also causes stress among fertile women.

Further, both fertile and infertile women who had infrequent intercourse(sometimes) during the fertile period were found to be more stressed. Even though the relationship between infrequent intercourse and stress might be bidirectional among both infertile and fertile women, but the reasons could be different. In the case of infertile women, when pregnancy does not happen even after trying multiple times over the years, the infertile couples tend to

**Table 7. Association of stress with other psychological variables among fertile and infertile women.**

| Psychological Variables | | Infertile | | Fertile | |
|---|---|---|---|---|---|
| Independent Variable | Dependent Variable | B coefficient (Standard error) | P-value | B coefficient (Standard error) | P- value |
| Stress | Depression | 0.144(0.044) | 0.001 | 0.155(0.04) | 0.001 |
| | Anxiety | 0.104(0.045) | 0.023 | -0.021(0.051) | 0.685 |

lose interest in sexual activities and develop stress, as a result of which, their sexual desires tend to reduce. Infertile couples often forget that their sexual relationship is also their natural need [28]. Thus, the stress due to infertility leads to infrequent or reduced sexual intercourse amongst infertile couples. Whereas, in the case of fertile women, stress could be due to high physical activity. Because of extreme physical activity and tiredness, women can get stressed, which further leads to infrequent sexual intercourse among the fertile couple. Moreover, less frequent sexual intercourse among both fertile and infertile women leads to the minimal release of endorphins, which is a natural mood booster, thus leading to higher levels of stress [29]. In the present study, linear regression analysis also revealed that infrequent intercourse during the fertile period increases stress among infertile women.

In reference to anxiety, the infertile women who were illiterate or had primary education were found to be more anxious. Traditionally, illiterate or less educated women belonging to the middle or lower class are typically involved in domestic tasks and are homemakers, so household chores, economic dependency, pressure to conceive from the husband and in-laws may lead to lower self-esteem and self-confidence in them which might give rise to anxiety. A study by Kaur et al. (2021) also reported that the prevalence of anxiety was found to be higher among the illiterate and homemakers compared to working women in India [30].

The infertile women who were physically inactive or only slightly active were more prone to anxiety. These women might be less active due to the underlying causes of infertility such as pain due to endometriosis [31], painful menses, ovarian cysts, etc. [32], which in turn might be leading to anxiety among them. Studies have also demonstrated that people having a sedentary life or those who were less physically active were found to be more anxious [33, 34].

Also, infertile women who got married at an early age were found to be more anxious. Sezgin & Punamaki (2020) also found that the early age at marriage is associated with worse mental health and poor wellbeing [35]. The women who were married young are deprived of their right to make vital decisions about their sexual health and wellbeing. In addition, owing to their infertility, they have to deal with various consequences such as violence, abuse, etc., which in turn might lead to anxiety among them [36, 37]. Again, the women who were infertile for more than 5 years were found to be more anxious. A study by Mirghafourvand et al (2016) also reported that anxiety was the most common after 4–6 years of infertility and especially severe anxiety could be found in those who had infertility for 7–9 years [38]. In the present study, the linear regression also revealed that the higher duration of infertility (6–10 years) is causing anxiety among infertile women.

Moreover, the fertile women below the age of 25 years were found to be more anxious. A study by Rebeca & Rebecca (2003) revealed that the younger age group of Indian women, i.e., below 25 years of age, were more vulnerable to anxiety and had less ability to deal with responsibilities, early pregnancy, and pregnancy-related mortality and morbidity [39]. It causes psychological issues like stress and anxiety in younger women as compared to older women [40]. The fertile women who were having undisturbed menstruation but heavy bleeding were found to be more anxious. Anxiety among fertile women could be due to pre-menstrual syndrome (PMS), which happens before every period, which in turn is associated with mood regulation and causes anxiety among women during or before menstruation [41]. The heavy bleeding or menorrhagia in fertile women could be due to fibroid, hormonal imbalance, or ovulatory dysfunction. A few studies have also reported that menorrhagia causes psychological problems especially anxiety in women [42]. In the present study, the relationship between heavy bleeding and anxiety is also found after regression analysis among the fertile women.

In reference to depression, infertile women who used to oversleep (> 8 hours) were more depressed. Oversleeping among infertile women could be due to the depression induced by infertility. Studies have found out that depression may contribute to sleep-related issues and people who have experienced depression can have excessive sleepiness [43]. A major factor

that could cause oversleeping in a depressed person is an interruption to a person's circadian rhythm [44, 45]. The infertile women who had less bleeding were found to be more depressed, there is a possibility that the infertile women might be attributing the cause of their infertility to less bleeding during menstruation, which, in turn, may lead to depression among them. A lesser amount of bleeding is found to be causing depression among infertile women, as has been revealed in linear regression as well.

In addition, the fertile women belonging to the joint family were found to be more depressed. The joint families do not provide enough scope for women to develop qualities of adventure, self-determination, which adversely affects their individuality, originality, and creativity, and this can further cause depression among fertile women [46]. The fertile women who exercised sometimes were found to be more depressed. Depressed people have a lack of interest in working out because when a person is suffering from depression, exercise is the last thing they want to do. A study by Kandola et al. (2019) reported that people with low aerobic and muscular fitness levels are almost twice as likely to experience depression [47]. Thus, it can be concluded that the relationship between irregular or no exercise and depression is bidirectional. Also, infertile women who slept for 7–8 hours were found to be more depressed.

Moreover, fertile women who were married at an early age were found to be more depressed. Marriage at an early age leads to early conception and more physical and mental burden among fertile women. It leads to greater psychological traumas such as loss of self-confidence. Among girls who were receiving education, early marriage tends to impede their education, resulting in severe depression [35]. Fertile women having infrequent (sometimes) intercourse were found to be more depressed and it could be because of less release of endorphins, which is a natural mood booster (released during intercourse), which in turn might be leading to depression [48]. Also, depression might be leading to infrequent intercourse. The neurotransmitters of the brain that trigger more blood flow to the sex organs don't function properly in depressed people and this, in turn, leads to loss of desire in sexual activity [49]. Finally, fertile women having normal bleeding during menstruation were found to be more depressed. Depression among women with normal bleeding might be due to premenstrual symptoms (PMS). Severe symptoms of PMS can also cause depression among women [50].

In sum, the results of the present study reveal that it is stress that in turn leads to depression as well as anxiety among infertile women. An earlier study also reported that chronic stress can lead to depression and anxiety in both infertile women and men and the depressive disorder often occurs together with anxiety in patients, the cause for both of these are strongly linked to stress [19]. On the other hand, stress only leads to depression and not anxiety among fertile women. The findings are in line with previous research which revealed that long-term stress or chronic stress can contribute to a mood disorder and depression among females [51].

Finally, in terms of a biological pathway, it is observed that stress leads to anxiety which in turn leads to depression [16]. However, it was found in the present study that stress is leading to both anxiety and depression among infertile women but only to depression in fertile women. These findings are in contradiction to the biological pathway, and need further investigation. Although the present study highlights the prevalence of various common mental disorders among infertile women independently with respect to demographic variables, lifestyles variables, and reproductive trajectories, more research is warranted in this direction to have a greater conviction in our findings.

## Conclusion

The infertile women attending gynecology OPD for treatment need to be counseled regarding reproductive trajectories mainly in terms of less bleeding during menstruation and infrequent

intercourse during the fertile period. There is a need to strengthen the infrastructure to educate and eradicate the myths around infertility by providing necessary guidance and counseling to both husbands and wives and incorporating mental health screening and treatment in the routine care of infertile women in India. We should try to build an ecosystem where women are encouraged to start discussing infertility-related issues and concerns openly without any apprehensions.

## Acknowledgments

I am thankful to the staff of Lady Hardinge Medical College, New Delhi. We express our gratitude to all the participants who cooperated with the research team. I also thank Jyoti Mishra for the idea of the research and research team Shivani Tyagi & Ankita Chandola for carrying out fieldwork in the times of the pandemic.

## Author Contributions

**Conceptualization:** Apoorva Sharma, Chakraverti Mahajan.

**Data curation:** Navjot Kamboj, Apoorva Sharma, Sukriti Dhingra.

**Formal analysis:** Navjot Kamboj, Apoorva Sharma.

**Funding acquisition:** Manju Puri.

**Investigation:** Manju Puri.

**Methodology:** Navjot Kamboj, Kallur Nava Saraswathy.

**Supervision:** Kallur Nava Saraswathy, Nandita Babu, Manju Puri, Chakraverti Mahajan.

**Writing – original draft:** Navjot Kamboj, Kallur Nava Saraswathy, Sweta Prasad, Apoorva Sharma, Sukriti Dhingra.

**Writing – review & editing:** Kallur Nava Saraswathy, Nandita Babu, Manju Puri, Mohinder Pal Sachdeva, Chakraverti Mahajan.

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
