## [Decision Letter · Decision Letter 0]

22 Nov 2022

PONE-D-22-16063Women Infertility and common mental disorders: A cross-sectional study from North India.PLOS ONE

Dear Dr. Mahajan,

Thank you for submitting your manuscript to PLOS ONE. After careful consideration, we feel that it has merit but does not fully meet PLOS ONE’s publication criteria as it currently stands. Therefore, we invite you to submit a revised version of the manuscript that addresses the points raised during the review process.

kindly review your manuscript in terms of statistical analysis and conclusion drawn from it. conclusion needs more robust explanation and should add information to existing literature  

We look forward to receiving your revised manuscript.

Kind regards,

Sidrah Nausheen, FCPS

Academic Editor

PLOS ONE

Journal Requirements:

"The authors received no specific funding for the publication of this work." 

Reviewers' comments:

Reviewer's Responses to Questions

**Comments to the Author**

1. Is the manuscript technically sound, and do the data support the conclusions?

Reviewer #1: Yes

Reviewer #2: Partly

2. Has the statistical analysis been performed appropriately and rigorously? 

Reviewer #1: Yes

Reviewer #2: I Don't Know

3. Have the authors made all data underlying the findings in their manuscript fully available?

Reviewer #1: Yes

Reviewer #2: Yes

4. Is the manuscript presented in an intelligible fashion and written in standard English?

Reviewer #1: Yes

Reviewer #2: Yes

5. Review Comments to the Author

Reviewer #1: The manuscript is technically sound and data that supports the conclusions. statistical analysis is rigorous, with appropriate controls and sample sizes. however, dealing with stress, anxiety and depression for fertile and infertile couple in a same study made it complicated and drawing a solid conclusion is difficult.

the Data was not shared publicly because of ethical issues. however, the source of data access is mentioned.

Reviewer #2: Interesting topic.

Results that are not significant can just be presented in Tables. No need to write their description again as it unnecessarily becomes complicated and confusing

Statistical analysis needs to be referrred to appropriate statistician

Conclusions drawn need to be compared and contrasted with contemporary literature. Very few references regarding these have been given.

Many of the conclusions are contemplations by authors and comparison with other studies has not been done

6. PLOS authors have the option to publish the peer review history of their article (what does this mean?). If published, this will include your full peer review and any attached files.

Reviewer #1: **Yes: **Dr Iffat Ahmed

Reviewer #2: No

---

## [Author Response · Author response to Decision Letter 0]

12 Dec 2022

Response to Reviewers:

The authors heartly thank the reviewers for their valuables comments on the manuscript titled “Women Infertility and common mental disorders: A cross-sectional study from North India.” The comments given by authors will surely enhance the manuscript and the authors have tried their best to incorporate the appropriate suggestions.

1. Entire manuscript has been formatted according to the journal guidelines.

2. Abstract: As suggested the results section has been rephrased to avoid repetition.

3. Background: 

i. Page no. 3 first paragraph later has been deleted.

ii. Page no. 3 second last para “( i.e. Stress and depression and stress and anxiety) among infertile and fertile women” has been deleted due to repetition.

4. Methods: 

i. Page no. 4 1st line Suchita has been changed to “Sucheta”

ii. Page no. 4 1st para “Both Infertile and fertile women were recruited based on strict inclusion and exclusion criteria.” has been deleted.

5. Results:

i. Page no. 5 1st heading “scores” is deleted.

ii. Page no. 7 last and 2nd last para has been deleted as suggested by the reviewers.

iii. Page no. 13 paragraphs has been deleted as suggested by the reviewers.

iv. Page no. 19 first para has been deleted as suggested by the reviewers.

v. Page no. 23-25 “units” has been changed to times. 

Reviewers comment: How was this calculated?

vi. Page 23- 25: This was calculated regression analysis. As, regression analysis depicts the relationship between two variables. It is applied in scenarios where the change in the value of the independent variable causes changes in the value of the dependent variable.

6. Discussion:

Reviewers comment: Comparison with other literature

i. References has been added wherever possible as suggested by the reviewers.

ii. Reference numbering have been changed

7. References:

i. References have been added wherever required.

ii. Formatting and numbering has been changed.

---

## [Editor Report · Decision Letter 1]

20 Dec 2022

Women Infertility and common mental disorders: A cross-sectional study from North India.

PONE-D-22-16063R1

Dear Dr. Chakraverti Mahajan,

We’re pleased to inform you that your manuscript has been judged scientifically suitable for publication and will be formally accepted for publication once it meets all outstanding technical requirements.

Kind regards,

Sidrah Nausheen, FCPS

Academic Editor

PLOS ONE
---

## [Editor Report · Acceptance letter]

26 Dec 2022

PONE-D-22-16063R1 

Women Infertility and common mental disorders: A cross-sectional study from North India. 

Dear Dr. Mahajan:

I'm pleased to inform you that your manuscript has been deemed suitable for publication in PLOS ONE. Congratulations! Your manuscript is now with our production department. 

Kind regards, 

on behalf of

Dr. Sidrah Nausheen 

Academic Editor

PLOS ONE